# SceneExpander: Real-Time Scene Synthesis for Interactive Floor Plan Editing

**Shao-Kui Zhang**
Tsinghua University
Beijing, China
shaokui@tsinghua.edu.cn

**Junkai Huang**
Tsinghua University
Beijing, China
huangjk21@mails.tsinghua.edu.cn

**Liang Yue**
Tsinghua University
Beijing, China
yuel22@mails.tsinghua.edu.cn

**Jia-Tong Zhang**
Tsinghua University
Beijing, China
2792504092@qq.com

**Jia-Hong Liu**
Tsinghua University
Beijing, China
liujiaho23@mails.tsinghua.edu.cn

**Yu-Kun Lai**
Cardiff University
Cardiff, United Kingdom
LaiY4@cardiff.ac.uk

**Song-Hai Zhang**[*]
Key Laboratory of Pervasive
Computing, Ministry of Education &
Tsinghua University
Beijing, China
shz@tsinghua.edu.cn

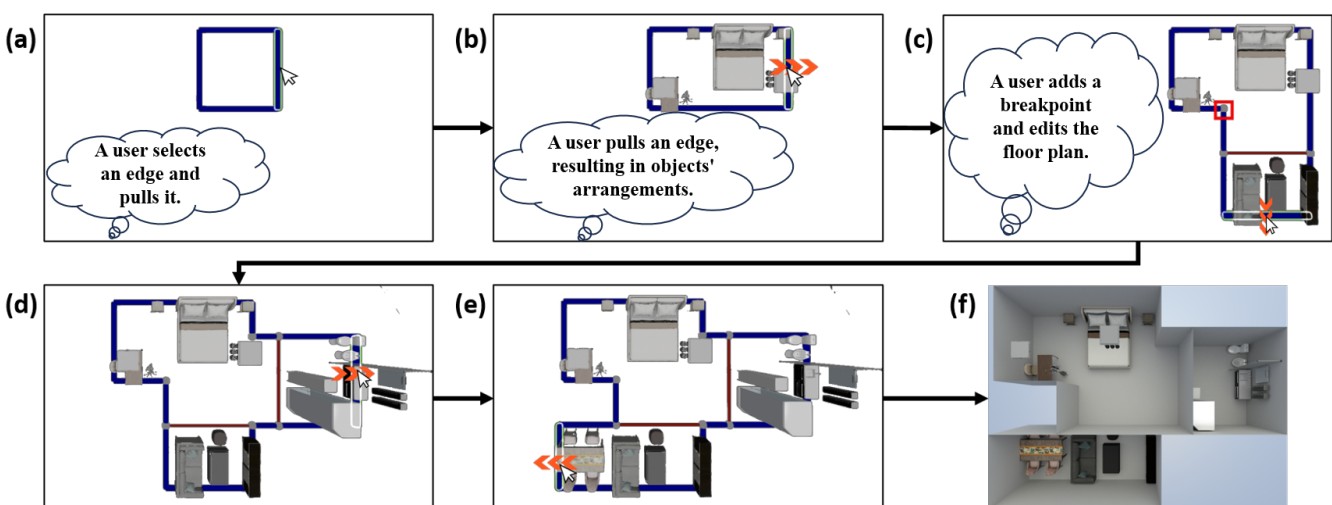

**Figure 1: We introduce a framework for synthesizing plausible scenes over interactively edited floor plans. Given an initial scene, a user can select an edge (wall) of it (a). The user can then expand (or shrink) the room shape, and our framework automatically adds (or removes) objects and continuously arranges objects (b). Users can add breakpoints (Red Box) to edit diverse floor plans (c). If a room is sufficiently large, our framework splits the room, adds new objects and arranges the existing objects (c, d, and e). Through a few operations, a floor plan that matches the user preference and contains plausible object arrangements is synthesized (f). Please see the supplementary video for interactive demos.**

[*]Corresponding Author.

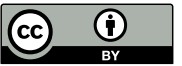

*MM '24, October 28-November 1, 2024, Melbourne, VIC, Australia*
© 2024 Copyright held by the owner/author(s).
ACM ISBN 979-8-4007-0686-8/24/10
https://doi.org/10.1145/3664647.3680798

## Abstract

Scene synthesis has gained significant attention recently, and interactive scene synthesis focuses on yielding scenes according to user preferences. Existing literature either generates floor plans or scenes according to the floor plans. The system proposed in this paper generates scenes over floor plans in real-time. Given an initial scene, the only interaction a user needs is changing the room shapes. Our framework splits/merges rooms and adds/rearranges/removes objects for each transient moment during interactions. A systematic

pipeline achieves our framework by compressing objects' arrangements over modified room shapes in a transient moment, thus enabling real-time performances. We also propose elastic boxes that indicate how objects should be arranged according to their continuously changed contexts, such as room shapes and other objects. Through a few interactions, a floor plan filled with object layouts is generated concerning user preferences on floor plans and object layouts according to floor plans. Experiments show that our framework is efficient at user interactions and plausible for synthesizing 3D scenes[1].

## CCS Concepts

• **Computing methodologies → Computer graphics**.

## Keywords

Scene Synthesis, Spatial Inference, User Interaction

**ACM Reference Format:**
Shao-Kui Zhang, Junkai Huang, Liang Yue, Jia-Tong Zhang, Jia-Hong Liu, Yu-Kun Lai, and Song-Hai Zhang. 2024. SceneExpander: Real-Time Scene Synthesis for Interactive Floor Plan Editing. In *Proceedings of the 32nd ACM International Conference on Multimedia (MM '24), October 28-November 1, 2024, Melbourne, VIC, Australia.* ACM, New York, NY, USA, 9 pages. https://doi.org/10.1145/3664647.3680798

## 1 Introduction

Research has emphasized the importance of synthesizing 3D scenes [18]. The plausibility of generating 3D scenes has significantly improved in the past decades [19]. Scene synthesis usually selects a set of objects and plausibly arranges them in a given floor plan [26]. The object arrangements should match the learned spatial relation priors and the strict constraints of the floor plan [5, 27, 29], e.g., the number of objects should fit the floor plan's capacity.

To synthesize scenes with user preferences, another branch of scene synthesis emphasizes interactively crafting scenes, i.e., controlling the process of scene synthesis. Through cursors [28] (e.g., using a mouse), texts [23], panels [25], etc., users can specify selections/translations/rotations for a few objects, and the frameworks arrange involved objects subsequently in each interactive session. For example, Clutterpalette [24] requires a user to select a position in the 3D scene for each interactive session. An object appears concerning existing objects and supports in the selected position.

However, existing literature mainly interprets user preferences as "selections and transformations" of objects, which is practical for users already satisfied with the provided floor plan. On the contrary, if users have not decided on a specific floor plan, they may repeatedly choose/modify a floor plan, interactively synthesize an object layout and decide to change the floor plan again, even if they only want to make minor tweaks to the floor plan. On the other hand, users may not know if the floor plan is ideal until they see how the objects can be arranged accordingly. To interactively synthesize a single floor plan with object layouts, a user has to conduct the synthesis process time after time for different floor plans. Interior designers typically face considerable floor plans. Therefore, the

challenge is to concurrently control floor plans and objects during the synthesis process, i.e., in the early stages of interior design, both the floor plan and objects' layouts can be provided simultaneously.

This paper introduces an interactive scene synthesis framework, where the interactive units are no longer objects but "room shapes" (floor plans). As shown in Figure 1, a user edits the floor plan for each interactive session, resulting in an expanded/cropped floor plan. Our framework judges the new floor plan and infers (a) rearrangement of objects, (b) splitting/merging rooms and (c) adding/removing objects. This entire inference process runs in real-time. Objects and rooms are dynamically and gradually changed by continuously adjusting the floor plan. Objects are continuously arranged along with changing floor plans at every transient moment, as shown in Figure 2. Users can stop when they are satisfied with the synthesized object layouts.

Our framework is executed iteratively for every transient moment. An algorithm pipeline is proposed to catch the changes in floor plans and decide if we should divide rooms, how we should divide rooms, how we should rearrange objects, and what objects should be added/removed (Section 3). The pipeline includes generating division plans, which help divide existing rooms so the floor plan is gradually enriched with more rooms. Our framework proposes multiple division plans for each transient moment, so it subsequently evaluates them and chooses the best or none of them (Section 4). If a room is enlarged, shrunk or divided, the pipeline may add more objects, remove objects that no longer fit the room, or arrange existing objects. We propose "Elastic Boxes" representing groups of objects for layout adjustment, as shown in Figure 2, to address the above operations (Section 5).

To our knowledge, we are the first to investigate interactively synthesizing scenes over editing floor plans. Our work makes the following contributions:

- We present an interactive scene synthesis framework that adds/rearranges/removes objects and splits/merges rooms over continuous changes of floor plans.
- We propose a system pipeline that detects room shape changes, executes room divisions, dispatches object groups and arranges objects in a transient moment of editing floor plans.
- We propose elastic boxes, which help spatially fit object groups to the continuous floor plan changes, thus plausibly arranging objects at any moment.

Note that our framework still arranges objects similar to typical synthesis frameworks, e.g., [24, 28]. Our framework does not generate floor plans [3, 6, 8, 13, 21] (Section 2.3). Instead, the floor plans are input from users and are interactively controllable.

## 2 Related Work

### 2.1 Interactive Scene Synthesis

Scene synthesis has been developed significantly in recent decades [26]. Researchers first investigate automatic scene synthesis, i.e., automatic selection and arrangement of objects in a room [5, 16, 20, 22, 27]. However, it is hard for this technique to generate scenes according to user preferences, e.g., a bedroom containing a double bed adjacent to the window. Therefore, interactive scene synthesis aims to control the scene synthesis process [28].

---

[1]Please refer to a supplementary video for an overview of our framework. The source code is publicly available at
https://github.com/Shao-Kui/3DScenePlatform#sceneexpander.

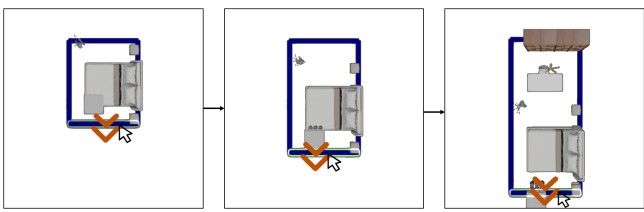

**Figure 2: This paper introduces elastic boxes, which allow continuous arrangement of objects according to the floor plan changes at any transient moment.**

Yu et al. [24] introduce a framework for placing small objects into scenes concerning supports, co-occurrence, etc. Zhang et al. [25] also place small objects but are concerned with the up/ down/ left/ right/ front/ back between objects. Merrell et al. [12] introduce a framework where users can specify constraints. The framework will generate several suggestions to be chosen. Zhang et al. [28] introduce a real-time framework for inserting objects into scenes. An object pops up appropriately for each transient moment while moving the cursor (e.g., with a mouse). Zhang et al. [30] allow editing a group of objects simultaneously, following the cursor. Additional attributes control objects in the same group.

Besides, literature introduces editing scenes passively, i.e., making users unaware of arranging objects. Liang et al. [10] rearrange objects in a workspace so the new layout can consume less commuting time. Zhang et al. [31] expand this to multi-users. They also calculate the optimal workspace layout and work allocation plan based on the individual work ability of employees, which can generate a more comprehensive scheduling plan. Facing virtual reality, Dong et al. [4] rearrange objects when users wear a Head Mounted Display to ensure they do not collide with obstacles. Liu et al. [11] propose a framework for arranging desktop items by studying user behaviors on the desktop, considering constraints such as object stacking and compactness.

The literature above allows editing objects with fixed floor plans. If a user is unsatisfied with the floor plan, e.g., needing spaces for more objects or new rooms, the user has to rearrange objects in the next floor plan from scratch. In our framework, an interactive unit is no longer an object or a group of objects. Users edit floor plans to indirectly manipulate objects, where the object layouts are synthesized according to the user-specified floor plans.

## 2.2 Fabrication

Existing methods also consider editing patterns inside 3D objects, e.g., rungs in the monkey bars. Funkhouser et al. [7] enable the creation of new 3D objects using existing objects. Schulz et al. [17] assemble components into a new object. Their method calculates the connections and alignments. Bokeloh et al. [1, 2] deform shapes while preserving object structures. They first search for discrete variations, such as component repetitions in an object. Then, the discrete variations are added or deleted when editing the objects. In contrast, Ovsjanikov et al. [14] explore 3D objects by continuously editing their components. A template model is extracted, deforming its components. Different objects are retrieved accordingly.

The above literature inspires us. We create new scenes using existing objects, where the components are groups of objects. However, indoor scenes usually do not need repeated patterns. Objects are inserted according to their functions.

## 2.3 Floor Plan Generation

Scene synthesis often assumes a floor plan is given, where objects are arranged. In contrast, other literature generates floor plans for scene synthesis [6]. Wu et al. [21] formulate neural networks to locate rooms, allocate areas for rooms, and assign room types. Nauata et al. [13] generate floor plans with a room graph, where each vertex indicates a room type. Hu et al. [8] generate floor plans with user constraints. They first generate graphs representing the relations of rooms. Then, the graph is decoded and aligned to yield floor plans. Deitke et al. [3] combine scene synthesis and floor plan generation into an automatic end-to-end framework.

The above methods can generate plausible floor plans, but they are end-to-end approaches, where a complete floor plan is generated given an input such as a graph. In contrast, our framework does not generate complete floor plans but explores continuous room divisions, i.e., a division is calculated for each transient moment. Hence, our framework seeks for controllable room division over editing floor plans. In other words, a floor plan is not "generated" but "adjusted" to adhere to user input.

## 3 Framework Pipeline

Figure 3 shows how our framework is executed for every transient moment, a minimal period detected by the computer. For example, hundreds of transient movements occur when a user moves a room shape's edge. The only input is a modified room shape. Our framework generates several division plans according to the changed room shape. A division plan indicates how to divide the room, and a room shape has multiple plans. Two evaluations subsequently decide whether the framework should divide the room by evaluating the current whole floor plan and the existing elastic boxes. Section 4 discusses generating and evaluating division plans. Whether a room should be divided has two branches. Each transient movement executes one of the branches and waits for the next transient moment, i.e., the next user interaction.

If the framework does not divide the room, the room shape is changed according to the interaction. Our framework then detects the elastic boxes that should be adjusted due to the modified room shape. For example, if an elastic box leans against a modified wall, the elastic box will be adjusted, i.e., translated according to the wall. Its adjacent elastic boxes are also involved in adjustments. This detection is recurred until no adjacent elastic boxes exist. "Shrinking" the room shape may remove one or more elastic boxes. After adjusting the elastic boxes, their objects are translated or rotated following their layout strategies, which suggest (1) object-object transformations, e.g., transforming a sofa concerning a coffee table, and (2) object-edge transformations, e.g., shrinking a living room may cause a sofa and a TV cabinet getting close to their coffee table. The object-object and object-edge transformations are maintained in a database. When all existing objects have been adjusted for this transient movement, our framework further attempts to add new elastic boxes by trying to find a free space. A new elastic box is

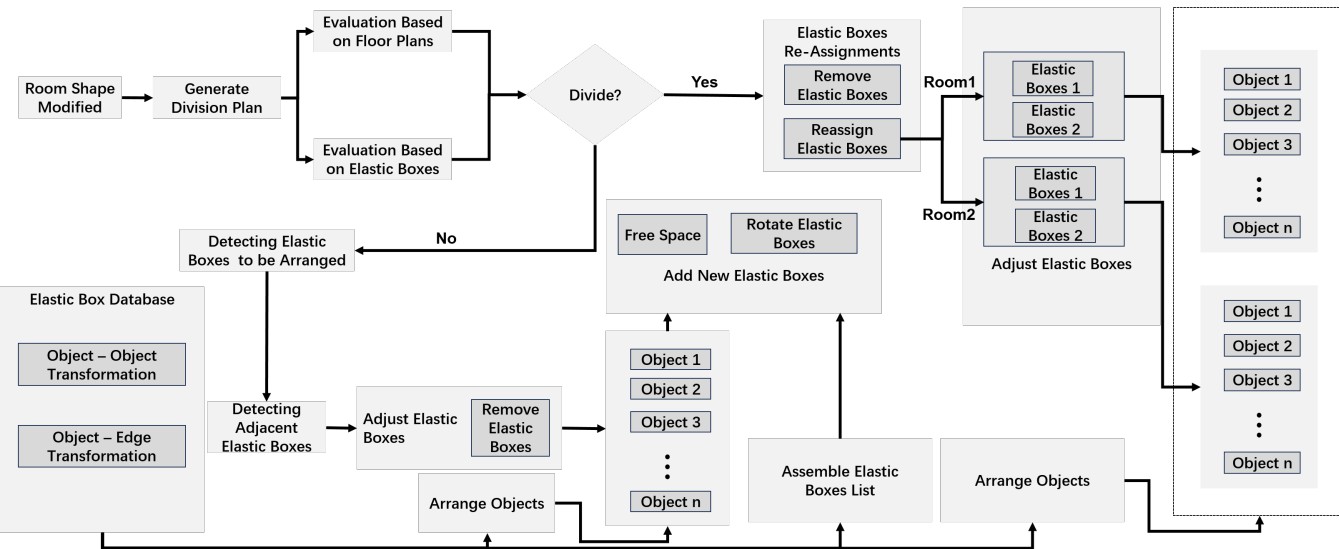

**Figure 3: At every transient moment, a room shape is modified, and we generate several division plans. A division plan indicates how a room should be divided according to the modified room shape. Two evaluations assess the division plans and decide if we should divide: (No): The modified room only changes its shape and the objects involved. We detect the elastic boxes that must be adjusted according to the new room shape. The objects inside elastic boxes are arranged according to their layout strategies (Section 5), which are stored in a database. (Yes): The modified room is split. The elastic boxes involved have been reassigned to the two new rooms. Then, the elastic boxes and their objects are adjusted and arranged.**

added and rotated according to a dependent wall if a free space exists.

If the framework should divide the room, a division plan is picked, and a new room is derived from the original room. Our framework examines each elastic box and reassigns it to one room, referring to room types and the ease of spatially moving the elastic box to another room. Every room has a type, e.g., bedroom, bathroom, etc. The two evaluations above also score the division plans, and the framework adopts the one with the highest score. A division plan contains the type for the new room indicating the initial elastic box, such as a "coffee table set" for living rooms. Since the room shapes are modified, the elastic boxes are adjusted, and their objects are arranged. Section 5 discusses the elastic boxes' mechanism and how we arrange elastic boxes' objects in detail.

Our framework supports merging two rooms into one if a room is shrunk. The merging logic aligns with the "Divide" branch in Figure 3. If, during the evaluation process, the framework decides that the room should be merged with another one, our framework will check each elastic box in the rooms to be merged, using the same evaluations above. The objects' arrangements follow how we utilize the elastic boxes above. We illustrate our framework to make our technical details self-contained, focusing on enlarging and dividing. Note that we allow users to choose their favoured room types, which can easily be added by giving tags to each room.

## 4 Room Divisions

Our framework splits the room by evaluating several division plans and choosing the best one. A division plan consists of two room shapes, $\mathbf{r}_1$ and $\mathbf{r}_2$, with room types $t_1$ and $t_2$. $\mathbf{r}_1$ and $\mathbf{r}_2$ are divided

from the original room. In each transient moment, only one room directly adjacent to the moving edge can be divided.

We generate division plans using a sampling method, as shown in Figure 4a, where an "inner edge" divides a room into two, and our method proposes a series of inner edges. Each inner edge differs from the previous division plan with a constant distance. The room adjacent to the moving edge is a new room that will be assigned a new room type, and the other room will keep the original type.

To evaluate a division plan, we define an evaluation function $\phi(\mathbf{r}, t)$ to measure how a room shape $\mathbf{r}$ and a room type $t$ are suitable, as shown in Equation 1, where $\beta(\mathbf{r})$ represents the number of the room $\mathbf{r}$'s edges. $\alpha(\mathbf{r})$ is the ratio of $\mathbf{r}$'s area to the area of its rectangular bounding box. The sum of $\beta(\cdot)$ and $\alpha(\cdot)$ evaluates the regularity of room shape $\mathbf{r}$. A high value of it indicates a regular room, such as a rectangle in Figure 4b. The exponential and logarithmic are applied, so the values' differences are significant.

$$\phi(\mathbf{r}, t) = - C_1 \exp(\beta(\mathbf{r})) + C_2 \ln(\alpha(\mathbf{r})) \\ + C_3 P_r(\mathbf{r}, t) + C_4 \min(1, \frac{\delta(\mathbf{r})}{\epsilon(t)}) \quad (1)$$

$P_r(\mathbf{r}, t)$ is the room area distribution function extracted from the RPLAN dataset[21]. For each room type $t$, the distribution varies. $\delta(\mathbf{r})$ is the minimal edge length of the room's bounding box, while $\epsilon(t)$ is a constant representing the desired minimal length of a room type $t$. $P_r(\cdot)$ and $\min(1, \frac{\delta(\mathbf{r})}{\epsilon(t)})$ ensure the room has proper space to accommodate the functionality of its type.

We also define another evaluation function, $\psi(\mathbf{c})$, to control the number of each type of room, as shown in Equation 2. Each entry

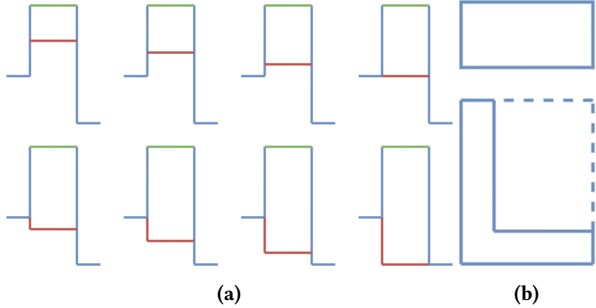

**(a)**                                              **(b)**

**Figure 4: (a): Generating Division Plan. Each of the sub-figures is a division plan. The green line represents the outer edge being moved, and the red lines represent the inner edges dividing the original room into two. (b): Two different rooms in the same area. The upper room has fewer edges and is closer to the bounding box; thus, it will have a higher value in the first two components of Equation 1.**

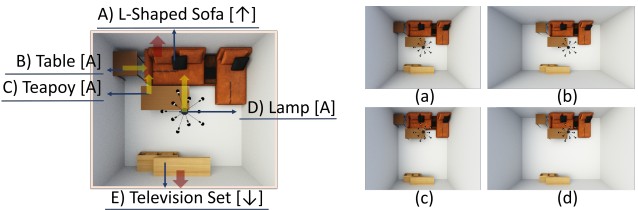

**Figure 5: LEFT: an elastic box and the involved objects' layout strategies. Our framework has two strategies. A yellow transparent arrow indicates another strategy, where the object is bound with another object, e.g., the teapoy is arranged concerning the relative transformations to the sofa. A red transparent arrow indicates a strategy where the object is bound with an edge (wall), i.e., the object is arranged concerning the relative transformations to the edge (wall). A solid blue arrow records an object's identifier, semantics and the edge/object to which it is bound. RIGHT: four examples of how the objects to the LEFT are arranged given different boundaries.**

of $\mathbf{c}$ is the number of a room type in the current editing floor plan, and $n$ is the number of room types, i.e., $\mathbf{c}$ is an $n$-dimensional vector.

$$\psi(\mathbf{c}) = \sum_{(\hat{\mathbf{c}},v) \in \mathcal{S}} \left( v \times \sum_{k=1}^{n} \left\{ \begin{array}{ll} \mathbf{P}_k \times \exp(\hat{\mathbf{c}}_k - \mathbf{c}_k) & \mathbf{c}_k \leq \hat{\mathbf{c}}_k \\ \mathbf{Q}_k \times \exp(\mathbf{c}_k - \hat{\mathbf{c}}_k) & \mathbf{c}_k > \hat{\mathbf{c}}_k \end{array} \right) \quad (2)$$

$\mathcal{S} = \{(\hat{\mathbf{c}}, v)\}$ is the distribution of room types extracted from RPLAN [21]. $\hat{\mathbf{c}}$ is a vector of room types similar to $\mathbf{c}$. $v$ is the frequency of $\hat{\mathbf{c}}$ in the dataset. $\mathbf{P}$ and $\mathbf{Q}$ are two positive constant vectors that measure if a new room should be added with a specific room type. Room types such as "living rooms" that are necessary but should not be many have large $\mathbf{P}_k$ and $\mathbf{Q}_k$. Necessary room types, such as "bedrooms", can be many and have a large $\mathbf{P}_k$ but a small $\mathbf{Q}_k$. Room types such as "storage" are unnecessary and have small

$\mathbf{P}_k$ and $\mathbf{Q}_k$. Then for each division plan $\{(\mathbf{r}_1, t_1), (\mathbf{r}_2, t_2)\}$, its overall evaluation $\sigma(\mathbf{r}_1, t_1, \mathbf{r}_2, t_2)$ is shown in Equation 3.

$$\sigma(\mathbf{r}_1, t_1, \mathbf{r}_2, t_2) = \min(\phi(\mathbf{r}_1, t_1), \phi(\mathbf{r}_2, t_2))$$
$$- C_t \times \psi(\tilde{\mathbf{c}}) - C_d \times \delta(\mathbf{r}_1, \mathbf{r}_2) \quad (3)$$

where $\tilde{\mathbf{c}}$ is the new room type count vector after new rooms are introduced, i.e., an entry of $\mathbf{c}$ increases. $\delta(\mathbf{r}_1, \mathbf{r}_2)$ is the number of elastic boxes removed, which will be explained in section 5. If a division plan eliminates too many elastic boxes, $\delta(\cdot)$ will be large. $C_t$ and $C_d$ are the trade-offs between the two evaluations.

We also evaluate the original room, i.e., $\phi(\mathbf{r}, t) - C_t \times \psi(\mathbf{c})$, which is compared with the evaluations yielded by various division plans. Suppose any of the division plans have a higher evaluation. In that case, the division plan with the highest evaluation is accepted, and the original room is divided, i.e., the "Yes" branch in Figure 3 and Section 3. Otherwise, the room is not divided, i.e., the "No" branch.

If a room is shrunk, we will determine the connectivity of the room $\mathbf{r}$ with the room from which it was divided. If they are connected and the evaluation of the merged room is higher than the evaluation of the two individual rooms, the two rooms are merged.

## 5 Elastic Boxes

An elastic box contains several objects, as shown in Figure 5. We use each object's "layout strategy" to calculate its transformation at every transient moment. A "layout strategy" is (1) an object's relative transformations concerning another object in the same elastic box (e.g., the yellow arrows in Figure 5) or (2) an object's relationship to a specific boundary of the elastic box (e.g., the red arrows in Figure 5). Each elastic box contains four boundaries. Each boundary may be adjacent to a room's edge (wall). Whenever the four boundaries are determined, the elastic box's objects are arranged according to our layout strategies, e.g., Figures 5(a-d).

An elastic box's semantics consist of its objects' semantics, indicating its function in an indoor scene. We utilize the Spatial And-Or Graph of Qi et al. [16] to infer elastic boxes given room types. Qi et al. [16] build a hierarchy to infer objects (children) from room types (parent). Our framework adds elastic boxes in between.

As shown in Figure 3, our framework contains three operations on elastic boxes. The first operation, "Evaluation Based on Elastic Boxes," evaluates division plans given existing elastic boxes. We attempt to assign elastic boxes from the original room to one of the new rooms. If the room's semantics fit the elastic box, we will pull the elastic box into the new room, where we first test the intersection of elastic boxes and rooms. If an elastic box and a room disjoint, the elastic box should not be moved into the room. Otherwise, we will check the intersection between the room's edges and the box's boundaries. The elastic box should be moved along the normal direction of the boundaries outside the room. Figure 6 illustrates more details. If the distance to move the elastic box is large, it should not be moved into the new room. The elastic box is removed if an elastic box should not be moved into both new rooms. Each elastic box from the original room is assigned to a new room or removed through this process.

The second operation is "Adjusting Elastic Boxes," which changes existing elastic boxes' boundaries and removes the elastic boxes that are too small after room shape modified. If an edge (wall) is moved, we enumerate each boundary adjacent to it, where the boundaries

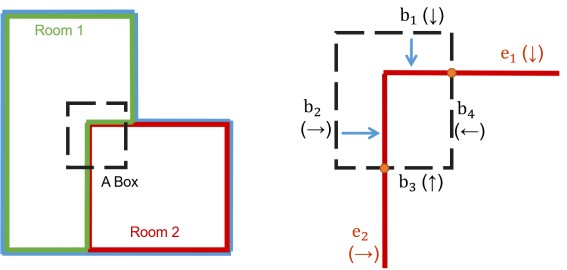

Figure 6: Evaluation Based on Elastic Boxes. LEFT: A division plan yields two new rooms (Green and Red). An elastic box (Black) intersects with the inner edge. RIGHT: The elastic box's four boundaries and two room edges are annotated with their indices ($b_i$ and $e_j$) and normal directions (Bracket). When we try to assign the elastic box to the new room (Red), we should pull the box along the normal direction of the boundaries (Blue Arrows). We check the intersections between the elastic box's boundaries and room edges, and edges $e_1$ and $e_2$ intersect with the box's boundaries. For example, we achieve the boundary $b_1$ facing the same direction as $e_1$, and calculate the distance between $e_1$ and $b_1$. The box is moved along the normal direction of $b_1$ for the distance.

are added to a pending set. For each boundary, we also check its opposite boundary's adjacencies, added to the set, as shown in Figure 7. If an edge is adjacent to a boundary, the boundary will not be moved, i.e., keeping leaning against the edge. If a boundary is adjacent to nothing, it will follow the edge (wall) dragged by the user. Then, all the boundaries in the set are adjusted according to their lengths and the distance caused by the user ($\Delta d$ in Figure 7).

The third operation is "Adding New Elastic Boxes," which detects free spaces that may hold a new elastic box, adds a new elastic box, and rotates it concerning a wall. The elastic box database initially proposes an elastic box whose semantics fit the current room. Our framework places the newly added elastic box next to each room corner and checks if it conflicts with other objects. If no conflict occurs at a corner, the box is added. Figure 8 shows an example.

## 6 Experiments

### 6.1 Implementation

We developed a web-based platform to hold the proposed framework, as shown in Figure 9. The platform supports editing or splitting room shapes. Users can freely move an edge or add breakpoints to it. At every transient moment, our platform detects a modified room shape, and our framework is executed so that objects are arranged. Our framework and platform are developed using Three.js.

The platform supports editing individual objects or objects of elastic boxes, such as typical industrial solutions or other interactive scene synthesis frameworks (see Section 6.2). Users can add, delete, move, rotate or scale individual objects. Therefore, we can combine our framework with other interactive frameworks, enabling interactive convenience and customization flexibility.

We utilize 3D-Front to extract the elastic boxes. Since 3D-Front mainly contains bedrooms and living rooms, we invite interior

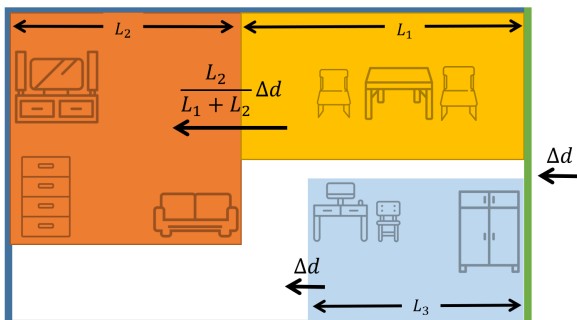

Figure 7: Adjusting Elastic Boxes. The right wall (Green) is dragged with distance $\Delta d$, a tiny value for every transient moment. This wall is adjacent to the right boundary of two elastic boxes (Blue and Yellow). The bottom elastic box's left boundary (Blue) has no adjacency, so it will be moved for $\Delta d$ following the right wall. The up-right elastic box's left boundary (Yellow) is adjacent to another elastic box (Orange), whose left boundary is adjacent to the left wall. This left boundary should not be moved, i.e. keep leaning against the left wall. The boundaries between the two elastic boxes (Blue and Yellow) should be moved according to their lengths, i.e. $L_1$ and $L_2$, so their moving distance is $\frac{L_2}{L_1+L_2}\Delta d$.

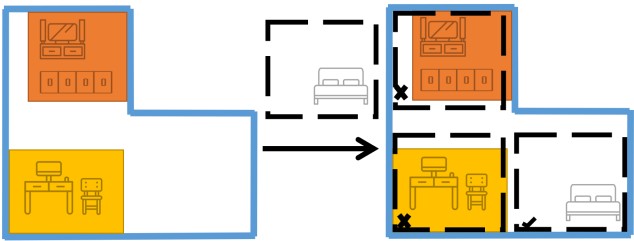

Figure 8: An example of "Adding New Elastic Boxes". For the current room (Blue) with two elastic boxes (Orange and Yellow), an elastic box (Black) is proposed. Then, our framework places it at each room corner until no conflict occurs, e.g., the box is placed at the lower right corner (Ticked).

designers to create objects and their arrangements for other room types, such as bathrooms. The elastic boxes are structurally stored in our database. We can tune the strategies, e.g., changing the relative positions required between objects/walls.

### 6.2 Usability and Efficiency

This experiment verifies our framework's usability and efficiency by comparisons. To our knowledge of interactive scene synthesis, there is no existing framework with exactly the same "interactive sessions" [30]. Our interactive session involves adding breakpoints and pulling edges (walls), indirectly arranging objects. Thus, we compare our framework with four baselines that interactively synthesize 3D scenes with their unique interactive sessions. We ensure the five frameworks (four baselines and ours) have the same input and output. The four baselines are:

**Table 1: Time Consumption. Each row refers to a framework. Each column refers to times consumed by a particular operation. Each cell contains an average time recorded and its standard deviation in brackets.**

| Framework | Navigate | Add | Remove | Translate | Rotate | Scale | Methods | Others | Total |
|---|---|---|---|---|---|---|---|---|---|
| Clutterpalette | 35.4 (33.0) | 16.7 (80.5) | 1.8 (2.6) | 29.2 (22.2) | 17.7 (13.5) | 1.1 (3.5) | 157.5 (85.1) | 105.2 (110.1) | 366.4 (206.9) |
| MageAdd | 27.7 (22.7) | 0.3 (1.5) | 1.6 (3.7) | 7.0 (11.6) | 4.4 (5.5) | 0.0 (0.0) | 167.6 (98.5) | 43.1 (38.8) | 251.6 (117.1) |
| SceneDirector | 20.1 (14.6) | 38.7 (98.0) | 2.8 (2.3) | 6.9 (8.0) | 3.7 (5.6) | 1.5 (3.5) | 56.1 (33.2) | 105.5 (123.8) | 235.4 (94.1) |
| Industrial | 65.6 (53.3) | 56.6 (21.1) | 1.6 (2.2) | 18.9 (12.7) | 16.6 (12.3) | 1.6 (4.8) | 0.0 (0.0) | 250.3 (98.5) | 411.2 (175.7) |
| SceneExpander | 17.2 (12.3) | 0.0 (0.0) | 0.4 (1.2) | 1.7 (3.1) | 1.0 (2.7) | 0.0 (0.0) | 53.9 (78.7) | 45.4 (58.6) | 119.6 (61.9) |

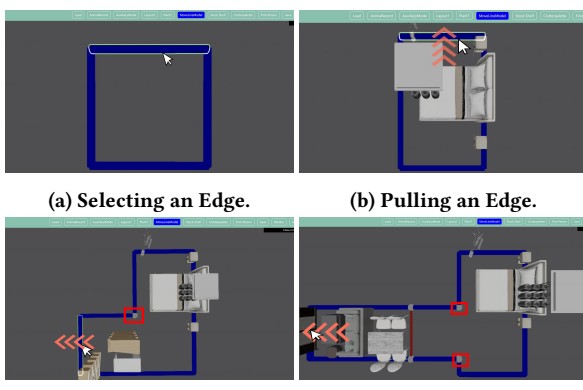

(a) Selecting an Edge.    (b) Pulling an Edge.

(c) Adding a Breakpoint.    (d) Adding two Breakpoints.

**Figure 9: The web-based platform for editing floor plans. Users can edit room shapes by selecting a wall (a) and pulling it (b). Users can add one (c) or two (d) breakpoints (Red Boxes) to split a wall, thus modifying walls' fragments.**

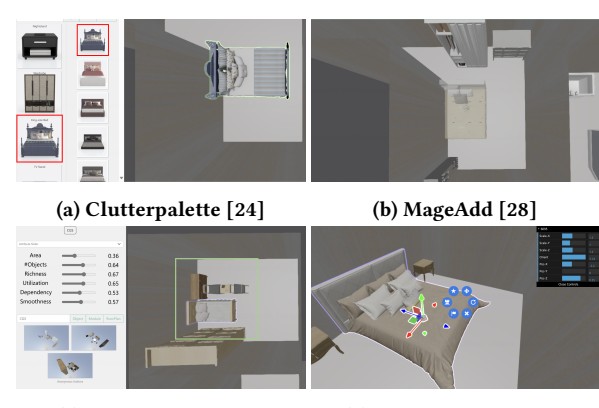

(a) Clutterpalette [24]    (b) MageAdd [28]

(c) SceneDirector [30]    (d) Industrial Software.

**Figure 10: The four baselines in addition to our framework. (a): Clutterpalette [24] adds objects (Red Boxes) according to their surrounding contexts. (b): MageAdd [28] explores potential objects that can be added, e.g., the translucent object in the scene. (c): SceneDirector [30] edits multiple objects concurrently (Green Box). (d): Typical industrial solutions are integrated into our platform.**

**Clutterpalette** [24]: Clutterpalette synthesizes a 3D scene by iteratively clicking positions, as shown in Figure 10a. A user clicks a position in each interactive session, and a set of objects appears on a UI based on the position's context. Selecting an object refines the recommended objects on the UI. When finding one, the user adds an object and continues another interactive session.

**MageAdd** [28]: MageAdd synthesizes a 3D scene by exploring potential objects that may appear in the scene, as shown in Figure 10b. In each interactive session, a user freely moves the cursor, and various objects may appear at the position pointed by the cursor. If the user finds a suitable object, the user can add the object, and the scene's contexts will change for more potential objects.

**SceneDirector** [30]: SceneDirector synthesizes a 3D scene by simultaneously editing multiple objects, as shown in Figure 10c. In each interactive session, a user selects an object, referred to as a dominant object according to [30]. Subsequently, a set of subordinate objects appears around the dominant one. When the user moves the dominant object, other objects follow it.

**Industrial Solution**: This denotes no "intelligent methods" involved, as shown in Figure 10d. Users follow typical industrial software such as Kujiale [9] and Planner 5D [15]. A user searches objects, adds objects, transforms existing objects or removes objects. Other "intelligent methods" sometimes suggest implausible layouts, so we enable industrial operations for all frameworks.

We invite 30 participants. Each participant uses the 5 frameworks to generate 5 floor plans with object layouts. Since our framework arranges objects over editing floor plans, to ensure the 5 floor plans yielded from the same participant have the same room shapes, a participant first uses our framework to edit an initial floor plan (with objects arranged by our framework). Then, she/he uses each baseline to synthesize a 3D scene using the same initial floor plan. All frameworks are implemented on the same 3D scene platform, as shown in Figure 9. The participants include professional interior designers, university students, freelancers, office workers, etc. All the participants were taught how to use our platform. A teaching video and a guide manual are available. They can freely try different frameworks until they are familiar with them. Each participant requires up to 1 hour to get familiar with the frameworks and approximately 1 hour to finish 5 floor plans. A technical staff member is nearby to answer technical questions and provide in-person instructions.

The participants include 16 males and 14 females, with an average age of 24.73 years old, with the youngest age 18 and the oldest 36. The participants have a variety of professional backgrounds. 16 of them are from architecture and interior design.

**Table 2: Interaction and Results. Each row refers to a framework. Each column refers to a rate in Section 6.3. Each cell contains an average rate recorded and its standard deviation.**

| Framework | Interaction | First-Person | Third-Party |
|---|---|---|---|
| Clutterpalette | 2.84 (0.99) | 2.68 (0.96) | 2.89 (1.04) |
| MageAdd | 3.16 (1.67) | 2.68 (1.06) | 2.89 (1.07) |
| SceneDirector | 3.58 (1.21) | 3.61 (1.31) | 3.28 (1.19) |
| Industrial | 2.74 (1.41) | 3.42 (1.31) | 2.85 (1.17) |
| SceneExpander | 3.59 (1.10) | 3.62 (0.89) | 3.31 (1.33) |

The participants were asked to interactively generate a floor plan until satisfied. For each generation, a participant can use only one of the above frameworks, while the "industrial Solution" has no intelligent method involved. We recorded the times consumed for the interactive synthetic processes. The time spent editing floor plans was not counted for the baselines. Because our framework was first executed, participants were more familiar with the baselines than ours, i.e., they will be more skilled in using the subsequent ones. Therefore, the time consumption was more advantageous to the baselines than ours. For different participants, the orders for using the four baselines were random.

Table 1 shows the average time spent on different user interactions, including (1) navigating the scene where users adjust views to interact with scenes from different perspectives. (2) adding objects where users drag searched results into scenes or duplicate objects. (3) removing objects. (4) translating objects. (5) rotating objects. (6) rescaling objects. (7) the frameworks' interactive sessions, such as simultaneously editing multiple objects [30]. (8) other time consumption, including elaborations, considerations, misoperations, etc, which are not trackable by our system. We add timers to every unit operation to ensure each consumed time is correctly recorded, so the systematic errors are negligible. Note that we still allow participants to use operations from "industrial solutions" to give them further preferences, and counting on this time, our framework still shows significant interactive time savings.

Each cell in Table 1 contains an average time and a standard deviation in brackets. Time is recorded in seconds. According to Table 1, our framework significantly reduces the time required to craft a 3D scene. A Kruskal-Wallis H-Test shows significant statistical differences between our framework's total time and the baselines' total time, with the p-value close to 0.

This experiment can also be treated as a usability study of recent interactive scene synthesis frameworks. For example, though SceneDirector [30] helps quickly arrange a set of objects concurrently, it does not arrange other objects irrelevant to the set. Thus, users need to add other objects manually. In contrast, though MageAdd [28] can be slow, it saves time in adding objects. Further discussions of the above frameworks are beyond the scope of this paper. See a supplementary video for a qualitative comparison.

## 6.3 Plausibility and Satisfaction

This experiment verifies our framework's plausibility and satisfaction. We invited the same participants in Section 6.2 to rate

our framework and the baselines based on "interaction" and "first-person score". The interaction refers to how participants feel a sense of smoothness, fluency and ease while interacting with the frameworks, with 0 being "poor feeling" and 5 being "excellent experience". The first-person score refers to how participants feel a sense of plausibility, aesthetics, and preferences in the scene they crafted, with 0 being "terrible" and 5 being "fine art".

Another 13 participants are invited to rate a third-party score on the results generated from Section 6.2. The newly invited participants do not intersect with those in Section 6.2. Each participant is presented with 50 random questions. Each question contains a 3D scene generated by one of the five frameworks. The third-party scores refer to how newly recruited participants feel a sense of plausibility and aesthetics in the scene presented. The new participants include professional interior designers and university students knowledgeable about arts or interior design. The participants do not know the frameworks. The participants include 6 male and 7 female participants. Their average age is 24.46 years old, ranging from 20 to 35. 5 are from architecture and interior design.

Table 2 shows the results. The "interaction" indicates that our framework maintains interactive smoothness, fluency and ease but significantly improves the interactive efficiency (Section 6.2). The "first-person score" indicates that our framework can generate results meeting user preferences since interactive scene synthesis aims to synthesize scenes incorporating user preferences [28]. The "third-party score" indicates that our framework can generate a plausible and aesthetic scene as a synthetic framework. Thus, the objects' arrangements from their elastic boxes are reasonable.

## 7 Conclusion

This paper introduces an interactive framework for scene synthesis. Users can continuously edit room shapes in each interactive session, so objects are arranged accordingly. Nevertheless, several drawbacks remain for future improvements.

First, as a graphical system works, our framework suffers from loading many objects concurrently. Our current optimization is caching and multi-threading, which keeps the framework smoothly operating when loading geometries and textures. However, our framework can not "foresee" elastic boxes to appear. If the framework keeps loading a few elastic boxes (e.g., loading complex models or encountering bad network traffic), nothing shall appear, even though our framework decides to add more objects. We should calculate elastic boxes' occupations and objects' arrangements to improve our framework upon adding new elastic boxes. In contrast, our framework must load all the 3D resources before the occupations and arrangements.

Second, hard decorations still need to be addressed. Similar to other interactive synthetic frameworks [24, 28, 30], our focus is still on objects and floor plans are not interactively synthesized. A complete "home decoration" still needs connections between rooms, which may conversely affect how we arrange objects, e.g., objects should not block a door between two rooms. Pipes and wiring are also currently not addressed in typical interactive/automatic scene synthesis frameworks.

## Acknowledgments

This work was supported by the National Key Research and Development Program of China (No. 2023YFF0905104), the National Natural Science Foundation of China (No. 62132012, 62361146854) and Tsinghua-Tencent Joint Laboratory for Internet Innovation Technology.

Shao-Kui Zhang is funded by Shuimu Tsinghua Scholar Program (No. 2023SM061), China Postdoctoral Science Foundation (No. 2024M751696), Postdoctoral Fellowship Program of CPSF (No. GZB20230353), Tsinghua University Student Research Training (No. 2421T0278, 2421T0277, 2411T0372, 2411T0371) and Young Elite Scientists Sponsorship Program by BAST (No. BYESS2024242).

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
