# OpenReview forum: "SceneExpander: Real-Time Scene Synthesis for Interactive Floor Plan Editing"
_acmmm.org/ACMMM/2024/Conference — MM2024 Poster_

### Official Review · Reviewer_Sj2D · 2024-05-08

**Rating:** 3
**Confidence:** 3

**Summary:**

This paper introduces an interactive framework for scene synthesis. The framework proposed by the author supports users to continuously edit the room shape in each interaction session, and supports the generation of object objects according to the room shape for corresponding arrangement.

**Strengths:**

1.From the experiment, it seems that the author's platform has been set up very completely. 2.The illustrations in the paper are designed to be very intuitive and easy to understand. 3.The paper provides a good review of the literature in the relevant field and demonstrates the background and importance of the research. 4.The paper designed interactive behaviors for multiple scenarios, and conducted detailed comparisons and analyzes over time when compared with baseline methods.

**Limitations:**

1.It is an interesting topic to allow users to modify the layout independently and choose the layout they like. However, I do not think it is novel enough and I have some doubts about the motivations proposed by the author, that is, lines 135 to 149, and I am not sure whether the author has relevant demand basis.
2.The article mentioned that the room reasoning process has been ongoing, but often the actual room layout should also consider other related factors such as pipes and wiring. I wonder if this reasoning process takes this into account. In addition, users often make independent choices when considering room layout, that is, they want a certain space to be a bedroom or a kitchen. I don’t seem to see this room selection being taken into account in the editing settings. If without those considerations, this work seems meaningless and need to be optimized.
3.I am curious about how the size of the elastic box is set. As mentioned in Figure 8, the author's framework places the newly added elastic boxes next to the corners of each room. Although this strategy can ensure that there is no obstruction between the elastic boxes, taking the flex box in Figure 8 as an example, will the frame setting in which the flex box where the sofa is located not intersect with other elastic boxes hinder more possible arrangements of the sofa itself in this layout?
4.I am not very clear about the author’s experimental design in the 6.2. If participants are asked to first use the author's framework to edit the initial floor plan, and then let other baseline methods use the initial floor plan to synthesize the scene, will it affect the participants' familiarity with the author's framework and the familiarity with other baseline methods on this initial floor plan? My problem may come from my failure to understand the author's experimental design. Please describe and explain it in more detail.

**Suitability:**

2

---

### Official Review · Reviewer_g7y2 · 2024-05-20

**Rating:** 3
**Confidence:** 2

**Summary:**

This paper presents a scene synthesize system, named SceneExpander, with novel interactive editing support. Given an initial scene layout, the system allows users to expand, shrink, and adjust the scene layout with real-time updates of the generated layout map and rendered furniture. The proposed system is implemented as a real-time interactive system. Through real-world user studies, this paper showed that SceneExpander outperforms several related works in editing time consumption and result qualities.

**Strengths:**

- The proposed system provides natural and real-time feedback to the scene designer, resulting in significant total time consumption compared to related works
- The proposed system has a complete end-to-end implementation with great potential for real-world application impacts.

**Limitations:**

- Although promising results are confirming the impacts of the proposed new user interactions, the drag-and-drop design on its own is not new. This drag-and-drop interaction design is commonly used in the 2D designing process, the author should look into the difficulties of providing precise control of the edges in 3D space.
  - Leveraging contextual views, such as top-view orthogonal projection, should help ease the issue.
- If possible, the evaluation setup section should include basic demographic information of the user study participants, such as gender. The added information will help in judging whether the participants' feedback on generation quality would be biased.
- There are several minor issues related to the paper's writing and formatting:
  - Page 1, line 37: This paper generates scenes over floor plans in real time. -> The system proposed in this paper generates scenes over floor plans in real-time
  - Figure 3 should be moved to the previous page as a complementary visualization for the descriptions of system workflow in section 3.

**Suitability:**

2

---

### Official Review · Reviewer_jmhc · 2024-05-20

**Rating:** 4
**Confidence:** 3

**Summary:**

This paper proposes a new interactive scene synthesis method over floor plan editing for interactive scene synthesis tasks. Considering that traditional interactive scene synthesis methods mainly focus on the user's interaction with scene objects, while ignoring the user's editing needs for the original floor plan, this paper pioneered a scene synthesis interaction method based on floor plan editing, allowing users to directly edit the initial floor plan, and proposed the concept of elastic boxes to achieve the chain change of related objects during floor plan editing.

**Strengths:**

1. Compared with the traditional scene interaction synthesis method, this method introduces floor plan editing for the first time, and introduces the concept of elastic boxes on the basis of plan editing to achieve real-time control of floor plan editing and scene object changes at the same time.
2. This method makes up for the lack of traditional methods in user floor plan editing needs, making the entire design process more in line with user needs. The experiments listed in this article can well support this contribution.
3. Effective scene interaction synthesis methods can effectively improve the efficiency of interior design. The method proposed in this paper not only enables users to customize the original floor plan, but also designs elastic boxes and algorithms to adjust and control the transformation of corresponding objects, greatly improving the editing efficiency of the original floor plan.

**Limitations:**

1. Indirectly arranging objects in the scene during the plan editing process improves efficiency to a certain extent, but it also loses some customization flexibility compared to the previous object-based editing methods, such as the placement customization of a single object and the free splitting and combination of multiple objects.
2. I have some questions about the settings of the elastic boxes: 1) Can the initial elastic boxes be edited in the subsequent process, such as adding, deleting, and rearranging internal objects; 2) Can the layout strategy in the elastic box be customized in the subsequent editing process?
3. Some limitations of doors and windows are mentioned at the end of the paper.

**Suitability:**

3

---

### Official Review · Reviewer_KBmT · 2024-05-23

**Rating:** 2
**Confidence:** 3

**Summary:**

The authors proposed a floor plan editing framework. This work can achieve the interaction for building a floor plan including using the elastic boxes and modifying the floor plan using its floor plan modification pipeline. It can achieve a better satisfactory rates from users than SOTA works.

**Strengths:**

The paper proposed work is a good representation of designing the floor plan and could be helpful to in-room design. Their work also can achieve the interaction with users to be more flexible and customized. The design of the self-adjustable elastic box is interesting and should be useful for basic designs.

**Limitations:**

1. This paper is more suitable for Graphic or HCI area conferences, not very close to the multimedia area
2. The lack of detailed explanation of the framework pipeline, for example, This work indicates the framework can merge the two rooms while one room is shrunk, as indicated in figure 3, and the elastic box to use in designs. Should add more experiments based on these designs.
3. The experiments combined with user studies, but the represented results only compared the time consumption and two rating scores for plausible and satisfactory.  Suggest considering more questions and evaluation scenarios for user study.
4. The discussion and explanation of Table 2 is not clear enough. Suggest discussing more about results and discussion from different scenarios.
5. Section 6.1, named "Results and setup", assumes this subsection should include results and setting up details. I can only find some brief implementation details.

**Suitability:**

1

---

### Meta-Review · Area_Chair_G2kS · 2024-07-02

**Recommendation:** Accept (Poster)
**Confidence:** 5

**Metareview:**

The paper presents a novel method to synthesise 3D scene interactively.

Even if reviewers don not have strong concerns about contributions and experimental evaluations, one reviewer has highlighted that the topic might be more suitable for Graphic and HCI venues (the paper has indeed only 28 cited papers and only a couple of them are from multimedia-related venues). The text should be improved clarifying missing information such as user study demographic and clarifying the experimental design. Two reviewers have however increase their score from borderline/weak reject to borderline accept after rebuttal phase.

The paper can be a good candidate for a poster presentation.
Please, in the camera ready address the comments from reviewers (i..e, missing information on user study, clarifying experimental design).